# Evaluation of a Clinical Index as a Predictive Tool for Primary Ciliary Dyskinesia

**DOI:** 10.3390/diagnostics11061088

**Published:** 2021-06-14

**Authors:** Vendula Martinů, Lucie Bořek-Dohalská, Žofia Varényiová, Jiří Uhlík, Václav Čapek, Petr Pohunek, Václav Koucký

**Affiliations:** 1Department of Paediatrics, Second Faculty of Medicine, Charles University and Motol University Hospital, 150 06 Prague, Czech Republic; lucie.dohalska@fnmotol.cz (L.B.-D.); Zofia.Varenyiova@fnmotol.cz (Ž.V.); venca@ciconia.cz (V.Č.); petr.pohunek@lfmotol.cuni.cz (P.P.); Vaclav.Koucky@seznam.cz (V.K.); 2Department of Histology and Embryology, Second Faculty of Medicine, Charles University, 150 00 Prague, Czech Republic; jiri.uhlik@lfmotol.cuni.cz

**Keywords:** primary ciliary dyskinesia (PCD), clinical index (CI), predictive tools, primary ciliary dyskinesia rule (PICADAR), North America criteria defined clinical features (NA-CDCF), nasal nitric oxide (nNO)

## Abstract

Background: In primary ciliary dyskinesia (PCD) there is no single diagnostic test. Different predictive tools have been proposed to guide referral of high-risk patients for further diagnostic workup. We aimed to test clinical index (CI) on a large unselected cohort and compare its characteristics with other widely used tools—PICADAR and NA-CDCF. Methods: CI, PICADAR, and NA-CDCF scores were calculated in 1401 patients with suspected PCD referred to our center. Their predictive characteristics were analyzed using receiver operating characteristics (ROC) curves and compared to each other. Nasal nitric oxide (nNO) was measured in 569 patients older than 3 years. Results: PCD was diagnosed in 67 (4.8%) patients. CI, PICADAR, and NA-CDCF scores were higher in PCD than in nonPCD group (all *p* < 0.001). The area under the ROC curve (AUC) for CI was larger than for NA-CDCF (*p* = 0.005); AUC_PICADAR_ and AUC_NA-CDCF_ did not differ (*p* = 0.093). An overlap in signs and symptoms among tools was identified. PICADAR could not be assessed in 86 (6.1%) patients without chronic wet cough. For CI laterality or congenital heart defects assessment was not necessary. nNO further improved predictive power of all three tools. Conclusion: CI is a feasible predictive tool for PCD that may outperform PICADAR and NA-CFCD.

## 1. Introduction

Primary ciliary dyskinesia (PCD) is a rare genetic disorder characterized by the impaired structure and function of motile cilia, which are responsible for mucociliary clearance in the respiratory tract. Moreover, motile cilia play an important role in other organs and systems, such as the reproductive tract (sperm flagella and cilia of uterine tubes) or embryonic primitive node, which is responsible for body left–right axis establishment (its dysfunction leads to laterality defects, such as situs inversus). The early symptoms of PCD come mainly from the upper and lower respiratory tract [1]. They include respiratory distress syndrome (RDS) in term newborns, persistent rhinitis in older children, recurrent otitis and hearing problems, wet cough as a symptom of chronic bronchitis, and bronchiectasis [2].

The diagnosis of PCD is challenging, as the symptoms may be non-specific and subtle. There is currently no single diagnostic test conclusively confirming PCD. The disease itself is very heterogeneous with mutations described in more than 50 genes [1,3] leading to slightly different phenotypes. This results in significant under-diagnosis and eventually the delayed initiation of appropriate therapy, which may improve patients’ quality of life and prognosis [4]. Currently, a combination of various diagnostic approaches is used to diagnose PCD. They include nasal nitric oxide (nNO), high-speed video microscopy (HSVM), transmission electron microscopy (TEM), immunofluorescence, and genetic analysis [5,6,7,8]. These methods are technically demanding and require personnel with a high level of expertise, which limits their use to specialized centers only. Due to the low availability of these methods, simple and broadly available screening tools to select patients requiring referral to diagnostic centers are needed.

Various questionnaires evaluating the patient’s probability of suffering from PCD have been proposed to better guide their center referral. In 2012, a simple clinical index (CI) to assess the risk of having PCD was proposed at our department [9]. It comprises a seven-item questionnaire (Table 1). Subsequently, it was validated on a cohort of 352 patients with chronic respiratory symptoms suspected of PCD and referred for HSVM testing (PCD was confirmed in 31 of them by TEM or genetic testing) [10]. Later (2016), primary ciliary dyskinesia rule (PICADAR) was proposed [11]. It has been validated for patients with chronic wet cough only and is based on seven additional variables (situs abnormality, gestational age, neonatal chest symptoms, admission to a neonatal intensive care unit (NICU), congenital cardiac defects, rhinitis, and ear and hearing symptoms). Some of them are difficult to recall in adults [11] or children with insufficient history or may require further diagnostic testing (chest X-ray or echocardiography). Another tool was developed by Leigh et al. in 2016 [12] in North America and defines four clinical criteria—North America criteria defined clinical features (NA-CDCF—laterality defects, unexplained neonatal RDS, early-onset year-round nasal congestion, and early-onset year-round wet cough). Both PICADAR and NA-CDCF have been recently validated on external cohorts. The authors showed no significant differences in the performance of either tool [13].

The primary aim of this study was to test the performance of CI on an unselected group of patients referred to our tertiary center for HSVM with suspected PCD. Furthermore, we intended to compare the predictive power of CI to that of later, but more widely used tools—PICADAR and NA-CDCF. As a secondary aim, we investigated the additional value of nNO for improvements in sensitivity and specificity when combined with CI, PICADAR, and NA-CDCF, respectively.

## 2. Materials and Methods

### 2.1. Ethics

This study was a part of grant project No. NV19-07-00210 by the Ministry of Health, Czech Republic, and was approved by the local ethics committee at the University Hospital Motol. Patients or their legal representatives gave informed written consent to participation in this project.

### 2.2. Study Population

We enrolled patients with suspected PCD who were referred to our center for HSVM testing between 1 January 2012 and 31 December 2020. Generally, these were patients with recurrent respiratory infections, chronic suppurative lung disease, bronchiectasis, chronic upper airway secretion, or laterality defects. Children under 1 year of age at the time of examination were not eligible for the study as clinically relevant data for respective questionnaires cannot be fully evaluated or may have limited predictive value.

### 2.3. Clinical Data

In all patients referred to our center for diagnostic PCD workup, relevant PCD signs and symptoms were analyzed in a patient’s history by a physician experienced in pediatric pulmonology. The data were recorded in a structured form in the medical documentation as part of the clinical routine. The subsequent diagnostic process of PCD was performed according to the ERS guidelines [8].

Briefly, patients older than 3 years underwent nNO measurement using an electrochemical analyzer Niox Mino (Aerocrine AB, Solna, Sweden) or Niox Vero (Circassia) according to a standard protocol based on the 2005 ATS/ERS recommendation [14] and ERS guidelines for the diagnosis of PCD [8]. A tidal breathing technique or oral exhalation against resistance in sitting patients was applied. Nasal air was aspirated via a nasal olive probe inserted into one nostril using a passive sampling flow rate of 5 mL·s^−1^ (∼0.3 L·min^−1^). The results were expressed in parts per billion (ppb).

In the next step, patients were tested by HSVM using the Keyence Motion Analyzer Microscope VW-6000/5000 via nasal brushing. The ciliary beat frequency and ciliary movement pattern were analyzed, adhering to relevant recommendations [15,16]. When suspicion of secondary dyskinesia was raised on the first examination, patients were invited for repeated testing after at least 4–6 weeks when free of respiratory symptoms or after antibiotic treatment (ATB). Patients with a high probability of PCD (with pathological ciliary movement or immotile cilia) were referred for TEM and genetic testing.

TEM was performed from nasal brushings or endobronchial biopsy samples, which were processed according to the laboratory protocols consistent with [17,18]. The interpretation of the TEM findings was performed by a physician experienced in TEM who adhered to the international consensus guidelines [19].

Genetic testing was performed using next-generation sequencing—a panel of ciliopathies that includes 39 PCD genes (KAPA hyperPlus kit (Roche, Pleasanton, CA, USA)), using SeqCap EZ Prime Choise Probes (Roche, Pleasanton, CA, USA). Additionally, extensive intragenic rearrangements of the DNAH5 and DNAI1 genes were investigated by the MLPA method using SALSA MLPA Probemix P238 and P237 (MRC Holland, Amsterdam, The Netherlands).

A definitive diagnosis of PCD was established in patients with a clear ultrastructural defect (based on TEM), a disease-causing mutation, or a combination of both. Inconclusive cases were discussed by a multidisciplinary board and referred for more advanced techniques, such as cell cultures, immunofluorescence, or advanced genetic testing (e.g., whole-exome sequencing). Subsequently, the diagnosis of PCD was based on a clinician’s judgment after considering all the relevant data.

As part of this study, three different predictive tools for PCD (CI, PICADAR, and NA-CDCF) were analyzed with respect to the signs and symptoms present. Answers to all questions from each questionnaire (Table 2) were retrieved from the medical data records. The actual scores of the individual predictive tools were calculated in each patient, as defined in the original publications [9,11,12]. Furthermore, levels of nNO in ppb were recorded, if available. Finally, a new possible predictive index (CI_new13_) composed of all signs and symptoms present in all previously mentioned predictive tools was proposed, and its characteristics were tested.

### 2.4. Statistical Analysis

The predictive characteristics of CI, PICADAR, NA-CDCF, and CI_new13_ were analyzed using receiver operating characteristics (ROC) curves with optimal thresholds, and the areas under the ROC curves (AUC) and their confidence intervals were evaluated. ROC curves were compared among CI, PICADAR, NA-CDCF, CI_new13_, and their combinations with nNO using DeLong’s test. Multivariable prediction models were built using logistic regression. Numbers of females and males were compared using a χ^2^-test. The values of the respective predictive tools between PCD and non-PCD groups were compared using a Mann–Whitney U-test. The frequency of the respective signs and symptoms included in CI, PICADAR, and NA-CDCF was compared between PCD and non-PCD groups using a test of difference between two proportions. *p*-values less than 0.05 were considered statistically significant. Analyses were performed using the R statistical package, version 3.6.3 (R Core Team (2020) R: A language and environment for statistical computing. R Foundation for Statistical Computing, Vienna, Austria). 

## 3. Results

Between 1 January 2012 and 31 December 2020, 1834 patients with suspected PCD were referred to our tertiary center for a PCD diagnostic workup. Complete clinical data for an evaluation of CI, PICADAR, and NA-CDCF and parents’ consent to participation in the research were obtained from 1401 of them, including 798 (57.0%) males. The age spectrum of the enrolled patients is shown in Figure 1, with the median age at referral to our center being 6.1 years and the oldest patient being 70.9 years; the oldest patient with confirmed PCD was 63.3 years. Successful nNO measurement was available in 569 patients, the youngest being 3 years of age. PCD was diagnosed in 67 patients (including eight inconclusive cases, which were reclassified to PCD after a multidisciplinary board discussion), i.e., 4.8%. Their characteristics, including their anthropometrics, spirometry, TEM, and genetic findings, are shown in Table 3.

In the three predictive tools of interest (CI, PICADAR, and NA-CDCF), 13 different signs and symptoms were identified. Persistent year-round rhinitis was present in each tool. There were also four characteristics included in two of them (neonatal respiratory symptoms, early onset year-round wet cough, laterality defects, and chronic ear symptoms or hearing problems), as shown in Table 2. CI and NA-CDCF could be evaluated in all patients, while PICADAR could not be evaluated in the 86 patients who did not suffer from wet cough (“condition sine qua non”). Except for preterm birth (*p* = 0.825), all signs and symptoms included in the respective predictive tools (*n* = 12) were significantly more frequent in PCD patients than in non-PCD patients. Median calculated values of CI, PICADAR, and NA-CDCF were significantly higher in PCD than in non-PCD patients (for CI: 5 vs. 3, *p* < 0.001; for PICADAR: 7 vs. 3, *p* < 0.001; for NA-CDCF: 3 vs. 2, *p* < 0.001, see Figure 2).

The area under the ROC curve for CI (AUC_CI_ = 0.884) was significantly larger when compared to NA-CDCF (AUC_NA-CDCF_ = 0.814), *p* = 0.005. When compared to PICADAR (AUC_PICADAR_ = 0.832), the AUC_CI_ was also larger, but this difference was particularly close to the level of predefined statistical significance (*p* = 0.066). The performance of PICADAR and NA-CDCF did not differ significantly (AUC_PICADAR_ vs. AUC_NA-CDCF_, *p* = 0.093). The best predictive characteristics were achieved for CI = 4 (specificity: 72.5%; sensitivity: 88.1%), PICADAR = 6 (specificity 87.3%, sensitivity 64.2%), and NA-CDCF = 3 (specificity: 94.0%; sensitivity: 56.7%). The predictive characteristics of the respective tools could be significantly improved when combined with nNO measurements (if available). The individual ROC curves are shown in Figure 3, and their characteristics are listed in Table 4. Figure 4 shows the probability curves for the respective tools.

Finally, we collated all 13 individual characteristics from three predictive tools of interest. One point for each was assigned if the characteristic was present—0 if not. This was tested as a possible new predictive index for PCD (CI_new13_). Its AUC was 0.900, its specificity was 0.687, and its sensitivity was 0.955, where the best threshold was 6. Its performance could even be increased when combined with nNO (AUC_CInew13_ + nNO = 0.932, *p* = 0.024). However, CI_new13_ did not outperform CI (*p* = 0.115), but it did outperform both PICDAR (*p* < 0.001) and NA-CDCF (*p* < 0.001) (Figure 3, Table 4).

## 4. Discussion

This study provides an independent and external validation of three predictive tools for PCD—CI, PICADAR, and NA-CDCF—on a large patient cohort. They all showed reasonable predictive power for patients with a diagnosis of PCD with AUC values over 0.800. In our cohort, we reached similar AUCs to those found in original publications [10,11] and to those of a recent external validation by Palmas et al. [13], who compared the performance of PICADAR and NA-CDCF on a cohort of 211 patients, including 25 with PCD, in the year 2020. In contrast to previously mentioned studies, in our cohort, PCD accounted for only 4.8% of patients (in a study by Palmas et al. [13], it accounted for 11.8%; in a PICADAR validation group, it accounted for 51.0%). This shows that our cohort was truly “unpreselected” and thus more relevant for everyday PCD screening.

Although we detected an overlap among CI, PICADAR, and NA-CDCF with respect to signs and symptoms being included in the individual predictive tools, we could find differences in their performance, with CI being clearly superior to NA-CDCF and nearly significantly better than PICADAR. CI is also more universal, as PICADAR cannot be used in patients without chronic wet cough (in our cohort, this corresponded to 6.1% of patients referred for PCD diagnostic setup). CI is also easy to establish, since it does not require any specific examination, relying only on data about the patient’s history. In contrast, PICADAR and NA-CDCF require a more detailed diagnostic workup to detect, e.g., congenital heart defects or laterality defects. A similar situation is found with other recently proposed predictive tools [2]. Notably, CI has a lower specificity than PICADAR and NA-CDCF but a higher sensitivity. This makes CI more suitable for first-line screening of PCD. With an NPV > 99% at an optimal threshold of 4, the diagnosis of PCD in a number of patients may be excluded with reasonable probability. Thus, its use may decrease the need for tertiary center referral. The relatively low specificity may be “compensated” for when other diseases with PCD-like symptoms are excluded. Thus, we recommend performing basic examinations first (e.g., a sweat test, a basic immunological examination, and an otorhinolaryngological examination, including an assessment of adenoid tissue and gastroesophageal reflux and chronic aspirations) to exclude other diseases with similar symptomatology but a higher frequency in the population and relatively easier diagnostics [20]. A reduced number of patients referred to a specialized diagnostic center will prevent their overload and will reduce the cost of a complex PCD diagnostic process.

Of our PCD group, only 38.8% of patients had laterality defects (Table 3). This is lower than that reported by Lucas et al. [21]. The diagnostics of PCD in patients with a laterality defect is more straightforward, and their mean age at diagnosis is lower than in those without these defects [22]. As the CI does not rely on laterality defects (while PICADAR and NA-CDCF do), it is mainly useful in the diagnostic process of these tough patients, where the diagnosis is usually delayed.

We also investigated ways to increase the predictive power of the respective tools. Nasal NO is a simple, cheap, easily performed test [23,24]. Its measurement may be successful in children older than 2–3 years. When combined with any of the predictive tools, nNO can significantly increase their performance, reaching a specificity of over 92% and a sensitivity of ≈80%. This is an encouraging finding, as nNO may be broadly available. We also addressed the question of whether it may be reasonable to expand the number of signs and symptoms included in the predictive tool. The new hypothetical 13-item questionnaire could not outperform CI, but outperformed PICADAR and NA-CDCF. Based on this finding, we suggest that, rather than increasing the number of items in the questionnaire, it is more reasonable to precisely specify the signs and symptoms being included in the questionnaire, as stressed by Leigh et al. [12], or refine their respective combination and relative powers (the number of points assigned to the respective signs and symptoms (like in PICADAR)).

Our study has several limitations. As there is no gold standard in diagnosing PCD, we cannot be sure that we did not miss some individuals with PCD in our cohort. Indeed, there were eight individuals with inconclusive findings (TEM and genetics), and they were finally reclassified as having PCD based on a multidisciplinary discussion. This could slightly distort our AUC calculations. This was also a retrospective study with a post hoc calculation of PICADAR and NA-CDCF (some patients in our cohort were referred to us prior to their publication in 2016). However, based on our detailed clinical data, we could answer all the questions included in PICADAR and NA-CDCF in the majority of patients referred to our center (76.4%). Moreover, we intentionally excluded patients younger than 1 year (see above). Thus, our results are not applicable to this age category. We also acknowledge the limitation of the questionnaire-based predictive tools, which require precise definition of the signs and symptoms being evaluated. Any deviation from the original definition may lead to a serious distortion of predictive power. However, we still believe that predictive questionnaires may help to improve the diagnostic process of PCD. A correct evaluation of the symptoms, early indication of examination, and diagnosis are important for the early initiation of treatment and a good prognosis in PCD patients.

## 5. Conclusions

This study compared three predictive tools for the diagnosis of PCD—CI, PICADAR, and NA-CDCF—and showed the differences in their predictive characteristics (AUC). CI is clearly superior to NA-CDCF and nearly significantly better than PICADAR in predicting PCD. Moreover, CI is a universal and easy-to-use tool, since it does not require any specific examination, in contrast to PICADAR and NA-CDCF, which require a more detailed diagnostic workup to detect, e.g., a congenital heart defect or a laterality defect. This study supports application of CI in a clinical routine.

## Figures and Tables

**Figure 1 diagnostics-11-01088-f001:**
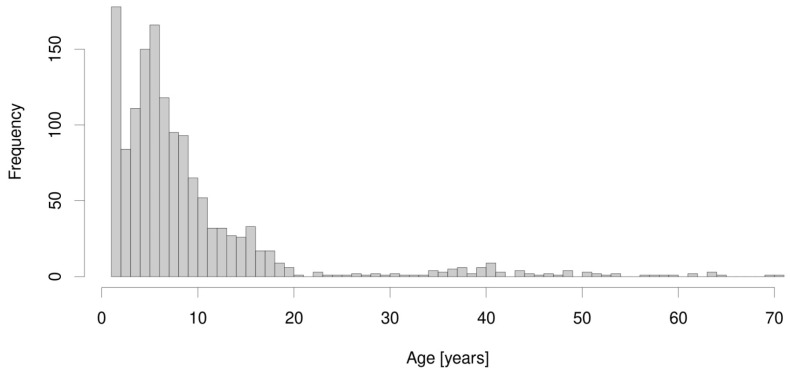
The age spectrum of the 1401 enrolled patients.

**Figure 2 diagnostics-11-01088-f002:**
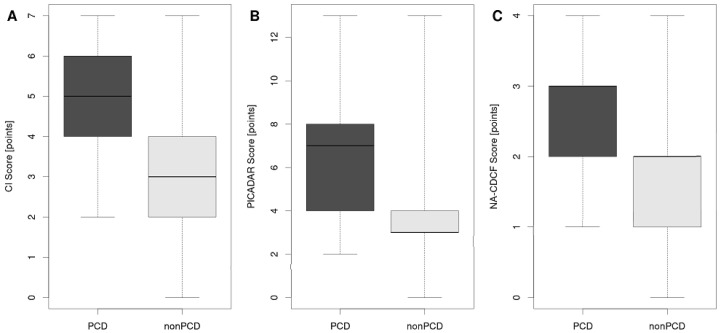
Median calculated values of CI (**A**), PICADAR (**B**), and NA-CDCF (**C**) in PCD and non-PCD patients. The box is drawn from the first to the third quartile with a horizontal line drawn in the middle to denote the median. Whiskers indicate the range of the data.

**Figure 3 diagnostics-11-01088-f003:**
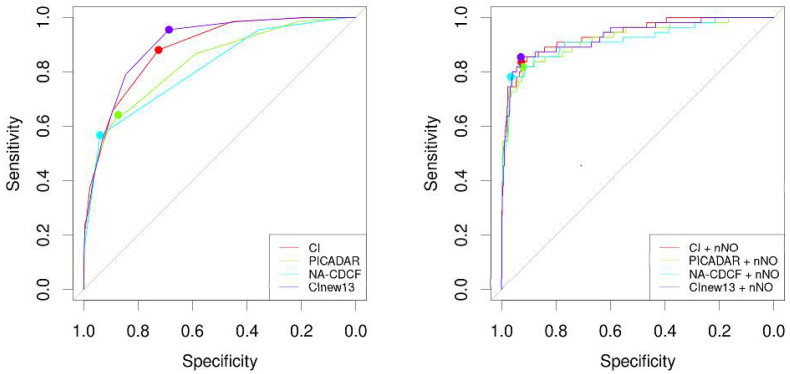
The individual ROC curves of CI, PICADAR, NA-CDCF, and CI_new13_ without or with nNO measurements.

**Figure 4 diagnostics-11-01088-f004:**
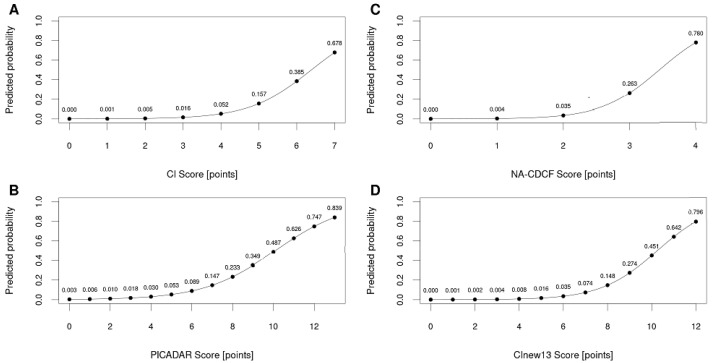
Probability curves for the respective tools—CI (**A**), PICADAR (**B**), NA-CDCF (**C**), and CI_new13_ (**D**).

**Table 1 diagnostics-11-01088-t001:** Clinical Index: Seven-item questionnaire [9,10].

**Clinical Index** **7-Item Questionnaire** **(Each YES = 1 Point)**	
	Did the child manifest with significant respiratory difficulties with breathing after birth?
	Did the child have rhinitis or excessive mucus production in the first 2 months of life?
	Did the child suffer from pneumonia?
	Did the child present with 3 or more episodes of bronchitis?
	Was the child treated for chronic secretoric otitis or suffered from >3 episodes of acute otitis?
	Does the child have a year-round nasal discharge or nasal obstruction?
	Was the child treated with antibiotics for acute upper respiratory tract infection >3 times?
**Overall score and risk**	**Proposed management**
0–1 point very low risk	PCD not suspected, focus on different causes of patient’s clinical symptoms, re-evaluate Clinical Index once a year. Refer the patient for PCD screening only if suspicion continues and other diagnoses were excluded
2 points low risk	
3 points medium risk	Exclude other causes of clinical symptoms and refer the patient for PCD screening.
4 points high risk	
5 + points very high risk	Probability of PCD is very high. Always refer the patient for HSVM.

**Table 2 diagnostics-11-01088-t002:** Overview of signs and symptoms present in the CI, PICADAR, and NA-CDCF questionnaire.

	CI	PICADAR	NA-CDCF
Pre-term vs. full term	−	+	−
Neonatal respiratory symptoms	+	+	−
Unexplained neonatal respiratory distress	−	−	+
Admission to a neonatal intensive care unit	−	+	−
Early-onset year-round wet cough	−	+	+
Rhinitis or nasal congestion in the first 2 months of life	+	−	+
Pneumonia in childhood	+	−	−
3 or more bronchitis episodes in childhood	+	−	−
Laterality defect	−	+	+
Congenital heart defect	−	+	−
Antibiotic therapy for rhinosinusitis > 3 times	+	−	−
Persistent year-round rhinitis	+	+	+
Chronic ear or hearing symptoms	+	+	−

**Table 3 diagnostics-11-01088-t003:** Characteristics of 67 PCD patients, including anthropometrics, spirometry, laterality defects, TEM, and genetic findings.

Median age at diagnosis (IQR)		6.1 (3.8; 9.8)	
Gender (% of males)		34 (50.74%)	
Height (z-score)		−0.03 ± 1.27	
Weight (z-score)		0.16 ± 1.60	
BMI (z-score)		0.07 ± 1.75	
FEV1 *		−2.10 ± 1.81	
FVC *		−1.99 ± 2.28	
FEV1/FVC *		−0.49 ± 1.40	
MEF25 *		−1.28 ± 1.55	
MMEF 25–75 *		−1.68 ± 1.20	
Laterality defects		26 (38.80%)	
TEM finding		Gene defect	
ODA	23 (34.32%)	DNAH5	13 19.4%)
		DNAI1	2 (2.98%)
		CCDC151	1 (1.49%)
		other	7 (10.44%)
ODA + IDA	21 (31.34%)	SPAG1	12 (17.91%)
		PIH1D3	1 (1.49%)
		DNAAF3	1 (1.49%)
		other	7 (10.44%)
MD + IDA	6 (8.95%)	CCDC39	3 (4.47%)
		CCDC40	1 (1.49%)
		other	2 (2.98%)
CP/TD	4 (5.97%)	RSPH4A	1 (1.49%)
		HYDIN	1 (1.49%)
		other	2 (2.98%)
Normal	13 (19.4%)	DNAH11	3 (4.47%)
		HYDIN	1 (1.49%)
		DRC1	1 (1.49%)
		other	8 (11.94%)

* Anthropometric and spirometric data are shown as mean ± SD. Gender and laterality defects are shown as absolute and percentage numbers. TEM, transmission electron microscopy, ODA, outer dynein arm defect, IDA, inner dynein arm defect, MD, microtubular disorganization, CP, central pair absence, TD, transposition defect.

**Table 4 diagnostics-11-01088-t004:** Predictive characteristics of the individual tools—CI, PICADAR, NA-CDCF, and CI_new13_ without or with nNO measurements.

Predictive Tool	AUC	LCL	UCL	Optimal Threshold	Specificity	Sensitivity	NPV	PPV
CI	0.8836	0.8479	0.9193	4	0.7249	0.9552	0.9918	0.1385
PICADAR	0.8319	0.7797	0.884	6	0.8734	0.6418	0.9785	0.2139
NA-CDCF	0.8141	0.7632	0.865	3	0.94	0.5672	0.9774	0.322
CI_new13_	0.9	0.86	0.93	6	0.68	0.95	0.99	0.13
CI + NO	0.9377	0.9012	0.9741	N.A.	0.928	0.8364	0.9815	0.5542
PICADAR + nNO	0.921	0.8763	0.9657	N.A.	0.9215	0.8182	0.9786	0.5357
NA-CDCF + nNO	0.9171	0.8679	0.9664	N.A.	0.9669	0.7818	0.9764	0.7167
CI_new13_ + nNO	0.9324	0.8918	0.9729	N.A.	0.93	0.8545	0.9835	0.5663

N.A.—not applicable, optimal threshold for nNO alone is 133.5 ppb. AUC, area under the ROC, LCL, lower control limit, UCL, upper control limit, NPV, negative predictive value, PPV, positive predictive value.

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
