# Peer review of "Evaluation of a Clinical Index as a Predictive Tool for Primary Ciliary Dyskinesia"

_diagnostics, 2021, doi:10.3390/diagnostics11061088_

Round 1

Reviewer 1 Report

In my opinion well written paper based on a Long Lasting (8 years) study. The only deficit is that comparison with a Gold Standard is lacking, but even histological diagnosis from nasal brushes and bronchial biopsy ist not always conclusive. Evaluation of anamnestic and non-invasive approaches towards diagnosis is therefore urgent. I think this study adds valuable new information

Author Response

Dear reviewer,

thank you for your comment. We are aware of that one gold standard test for PCD diagnostic does not exist, therefore we appreciate the opportunity to publish this anamnestic and non-invasive approach. 

Kind regards,

Vendula Martinu

Reviewer 2 Report

In line 18, the sentence 'patients older 3 years' doesn't seem to make sense.

Some abbreviations used like AUC and ATB may seem obvious but should be described earlier in the text.

The term akinetic is used, but I am not sure if it can be applied to cilia, the authors should double-check this

In table 3, 'other' has a cross, but the cross doesn't seem to be mentioned anywhere. This should be rectified (removed or explained)

Author Response

Dear reviewer,

thank you for your comments. Our revisions are marked up using the "Track Changes" to be easily viewed.
1. In line 18, the sentence 'patients older 3 years' doesn't seem to make sense - the text has been changed to "patients older than 3 years"

2. Some abbreviations used like AUC and ATB may seem obvious but should be described earlier in the text. - abbreviations have been explained in the text

3. The term akinetic is used, but I am not sure if it can be applied to cilia, the authors should double-check this - the term akinetic has been chanced to immotile

4. In table 3, 'other' has a cross, but the cross doesn't seem to be mentioned anywhere. This should be rectified (removed or explained) - crosses have been removed

We hope we have adequately revised the text according to your comments.

Kind regards,

Vendula Martinů